# HM-Chromanone Isolated from *Portulaca Oleracea* L. Protects INS-1 Pancreatic β Cells against Glucotoxicity-Induced Apoptosis

**DOI:** 10.3390/nu11020404

**Published:** 2019-02-14

**Authors:** Jae Eun Park, Youngwan Seo, Ji Sook Han

**Affiliations:** 1Department of Food Science and Nutrition, Pusan National University, Busan 46241, Korea; jaeeun5609@naver.com; 2Division of Marine Bioscience, Korea Maritime and Ocean University, Busan 49112, Korea; ywseo@hhu.ac.kr

**Keywords:** (*E*)-5-hydroxy-7-methoxy-3-(2′-hydroxybenzyl)-4-chromanone, *Portulaca oleracea*, INS-1 pancreatic β cell, glucotoxicity

## Abstract

In this study, we investigated whether (*E*)-5-hydroxy-7-methoxy-3-(2′-hydroxybenzyl)-4-chromanone, a homoisoflavonoid compound isolated from *Portulaca oleracea* L., protects INS-1 pancreatic β cells against glucotoxicity-induced apoptosis. Treatment with high glucose (30 mM) induced apoptosis in INS-1 pancreatic β cells; however, the level of cell viability was significantly increased by treatment with (*E*)-5-hydroxy-7-methoxy-3-(2′-hydroxybenzyl)-4-chromanone. Treatment with 10–20 µM of (*E*)-5-hydroxy-7-methoxy-3-(2′-hydroxybenzyl)-4-chromanone dose-dependently increased cell viability and significantly decreased the intracellular level of reactive oxygen species (ROS), thiobarbituric acid reactive substances (TBARS), and nitric oxide levels in INS-1 pancreatic β cells pretreated with high glucose. These effects were associated with increased anti-apoptotic Bcl-2 protein expression, while reducing pro-apoptotic Bax, cytochrome C, and caspase 9 protein expression. Treatment with (*E*)-5-hydroxy-7-methoxy-3-(2′-hydroxybenzyl)-4-chromanone reduced the apoptosis previously induced by high-level glucose-treatment, according to annexin V/propidium iodide staining. These results demonstrate that (*E*)-5-hydroxy-7-methoxy-3-(2′-hydroxybenzyl)-4-chromanone may be useful as a potential therapeutic agent to protect INS-1 pancreatic β cells against high glucose-induced apoptosis.

## 1. Introduction

Type 2 diabetes is characterized by a progressive decline in β-cell function and impaired insulin secretion [1,2,3]. Insulin secreted from pancreatic β-cells is an important factor in controlling blood glucose levels. Pancreatic β-cell dysfunction is a decisive factor in the pathogenesis of type 2 diabetes [4,5,6]. Pancreatic β-cells are sensitive to oxidation and are easily damaged by hyperglycemia-induced oxidative stress. Hyperglycemia promotes the production of reactive oxygen species (ROS), the constant generation of which leads to chronic oxidative stress over time. Pancreatic β-cells are damaged by oxidative stress, thereby reducing insulin secretion and ultimately leading to apoptotic cell death. Therefore, to diminish the hazard of pancreatic β-cell apoptosis, it is vital to identify methods able to reduce hyperglycemia-induced oxidative stress [7,8].

*Portulaca oleracea* is a herbaceous succulent annual plant belonging to the family *Portulacaceae* [9]. It is considered a functional food because of its richness in bioactive components, such as flavonoids, alkaloids, polysaccharides, fatty acids, terpenoids, and homoisoflavonoids [9,10,11]. Many studies have shown that it exhibits a wide range of therapeutic effects, including antioxidant, neuroprotective, and anti-diabetes effects [12,13,14]. 

The compound (*E*)-5-hydroxy-7-methoxy-3-(2′-hydroxybenzyl)-4-chromanone (HM-chromanone) is a homoisoflavonoid isolated from *P. oleracea* via bioassay-guided fractionation and HPLC. Homoisoflavonoids are an uncommon subclass of flavonoids that contain one additional carbon atom. Lin et al. classified homoisoflavonoids according to their carbon skeleton: sappanin-type (I), scillascillin-type (II), brazilin-type (III), caesalpin-type (IV), and protosappanin-type (V). HM-chromanone is a homoisoflavonoid in the form of a sappanin type with a 3-benzylchroman skeleton, where the benzopyran and the aromatic ring are connected via a single carbon. To date on, studies on the bioactivity of homoisoflavonoids have focused their anti-oxidant and cytotoxic effect [15,16]. Until now, the effects of HM-chromanone on pancreatic β-cell functions and cell recovery have not been previously reported. Therefore, in this study, we investigated the protective effects of HM-chromanone against INS-1 pancreatic β cell apoptosis induced by high glucose, and antidiabetic activities.

## 2. Materials and Methods 

### 2.1. Materials

The aerial part of *P. oleracea* plants were collected from Hongcheon Hyosung Food (Hongcheon Hyosung Food Inc., Gangwon, Hongcheon, Korea). The samples were washed three times using tap water to remove any salt, sand, and epiphyte, before carefully rinsing with fresh water. The samples were lyophilized and homogenized using a grinder (Shinhan Science & Technology Co., Kyunggi, Korea) prior to extraction. 

### 2.2. Extraction and Isolation

Dried *P. oleracea* powder (300 g) was extracted with decuple of methylene chloride (CH_2_Cl_2_) over 3 days at room temperature. The resulting extracts were filtered through Whatman No. 1 filter paper. The filtrate was then evaporated at 40 °C to obtain the CH_2_Cl_2_ extract (10.86 g). The extract was suspended in CH_2_Cl_2_, and the aqueous layer was partitioned with H_2_O. Next, the CH_2_Cl_2_ (14 g) extract was fractionated with *n*-hexane and 85% aqueous (aq.) methanol. The 85% aq. MeOH fraction (2.06 g) was separated into seven sub-fractions by RP18 column chromatography elution with stepwise gradient mixtures of 50% aq. MeOH, 60% aq. MeOH, 70% aq. MeOH, 80% aq. MeOH, 90% aq. MeOH, 100% MeOH, and 100% ethanol. The 80% aq. MeOH fraction (0.64 g) was further separated into 11 sub-fractions by silica column chromatography and eluting with stepwise gradient mixtures of CHCl_3_/MeOH (100% CHCl_3,_ 5, 10, 20, 30, 40, 50, 60, and 70% MeOH in CHCl_3_, 100% MeOH and 90% aq. MeOH). HM-chromanone was isolated by reversed-phase HPLC (ODS-A, 75% aq. MeOH) using 5% MeOH in the CHCl_3_ fraction (0.35 g). The structure of the compound was elucidated using a combination of spectroscopic methods, including HR/Mass, and ¹H and ¹³C NMR [17]. 

### 2.3. Cell Culture 

INS-1 pancreatic β cells were cultured in RPMI 1640 medium (Welgene, Gyeongsan, Gyeongsangbuk-do, Korea), supplemented with 10% of fetal bovine serum (FBS), streptomycin, and penicillin (100 units/mL and 100 μg/mL, respectively) and 50μM beta-mercaptoethanol at 37 °C in a humidified atmosphere containing 5% CO_2_.

### 2.4. Cell Viability Assay 

Cell viability was assessed using a colorimetric MTT (3-(4,5-dimethylthiazol-2-yl)-2,5-diphenyltetrazolium bromide) assay which measures the conversion of yellow MTT to purple formazan using mitochondrial enzymes. Cells (2 × 10^4^ cells/well) were pre-incubated with glucose (5.5 or 30 mM) in all the wells of a 96-well plate for 48 h, then incubated with 0, 1, 5, 10, or 20 µM HM-chromanone for 48 h. MTT solution (100 µL of 1 mg/mL) was added to each well of the 96-well plate and incubated for 4 h at 37 °C before the MTT-containing medium was removed. The formazan crystals in the viable cells were rendered soluble by adding 100 μL of dimethyl sulfoxide (Sigma, St. Louis, MO, USA). The absorbance of the contents of each well was determined at 540 nm using a microplate reader.

### 2.5. Assay of Intracellular ROS Levels

The intracellular ROS levels were measured using a dichlorofluorescein assay. The cell-permeable, non-fluorescent probe 2′,7′-dichlorodihydrofluorescein diacetate (DCF-DA) can be deacetylated in cells, where it quantitatively reacts with intracellular radicals (mainly hydrogen peroxide) and is converted into a fluorescent product, DCF, that is retained within the cells. DCF-DA was therefore used to evaluate the generation of ROS under oxidative stress. Cells (2 × 10^4^ cells/well) were pre-incubated with glucose (5.5 or 30 mM) in the wells of a 96-well plate for 48 h then incubated with 0, 1, 5, 10, or 20 µM HM-chromanone for 48 h. Then, the cells were washed with phosphate-buffered saline (PBS) and incubated with 5 μM DCF-DA for 30 min at room temperature. The fluorescence was measured using a fluorescence plate reader (LS-3B; Perkin-Elmer, Waltham, MA, USA).

### 2.6. Assay for Measurement of Lipid Peroxidation

Lipid peroxidation was measured based on the production of thiobarbituric acid reactive substances (TBARS). Cells (2 × 10^4^ cells/well) were pre-incubated with glucose (5.5 or 30 mM) in a 96-well plate for 48 h and then incubated with 0, 1, 5, 10, or 20 µM HM-chromanone for 48 h. Then, 200 μL of each medium supernatant was mixed with 400 μL of the TBARS solution. This mixture was boiled at 95 °C for 20 min. The absorbance at 532 nm was measured, and the TBARS concentration determined using a 1,1,3,3-tetraethoxypropane serial dilution standard curve. The TBARS values were expressed as equivalent nmoles of malondialdehyde (MDA).

### 2.7. Assay to Measure Nitric Oxide (NO) Levels

Cells (2 × 10^4^ cells/well) were pre-incubated with glucose (5.5 or 30 mM) in the wells of a 96-well plate for 48 h and then incubated with 0, 1, 5, 10, or 20 µM HM-chromanone for 48 h. The NO levels in the cell culture supernatant were measured using the Griess reaction. The cell culture supernatant (50 μL) was mixed with an equal volume of the Griess reagent (0.1% N-(1-naphthyl)-ethylenediamine and 1% sulfanilamide in 5% phosphoric acid), and the mixture was incubated at room temperature for 10 min. The absorbance was measured at 550 nm using a microplate absorbance reader. A series of known concentrations of sodium nitrite was used as a standard. 

### 2.8. Glucose-Stimulated Insulin Secretion (GSIS)

Cells (2 × 10^4^ cells/well) were pre-incubated with glucose (5.5 or 30 mM) in the wells of a 96-well plate for 48 h and then incubated with 0, 1, 5, 10, or 20 µM HM-chromanone for 48 h. Thereafter, the cell culture medium was carefully removed; the cells were washed with PBS, and fresh medium was added containing 3 mM glucose and 2% FBS. After 5 h, the cells were stimulated with Krebs–Ringer buffer (119 mM NaCl, 4.75 mM KCl, 2.54 mM CaCl_2_, 1.2 mM MgSO_4_, 1.2 mM KH_2_PO_4_, 5 mM NaHCO_3_, and 20 mM HEPES, pH 7.4) containing 5 or 25 mM glucose for 60 min at 37 °C, then the cell medium was collected for analysis of insulin secretion. Insulin secretion was determined using the rat/mouse insulin enzyme-linked immunosorbent assay (ELISA) kit (LINCO Research Inc., St. Charles, MO, USA).

### 2.9. Western Blot Analysis 

Cells (2 × 10^4^ cells/well) were pre-incubated with glucose (5.5 or 30 mM) in a 96-well plate for 48 h and then incubated with 0, 1, 5, 10, or 20 µM HM-chromanone for 48 h. Cell lysates were prepared using ice-cold lysis buffer containing 250 mM NaCl, 25 mM Tris–HCl (pH 7.5), 1% (*v*/*v*) NP-40, 1 mM dithiothreitol, 1 mM phenylmethylsulfonyl fluoride, and protease inhibitor cocktail (10 μg/mL aprotinin, 1 μg/mL leupeptin). Cell lysates were washed by centrifugation, and the protein concentrations were determined using a BCATM protein assay kit (Bio-Rad, Hercules, CA, USA). The lysates (30 μg protein) were electrophoresed on 10% sodium dodecyl sulfate-polyacrylamide gels, and the separated proteins were transferred onto nitrocellulose membranes. The membranes were incubated separately with antibodies against Bax, Bcl-2, cytochrome c, caspase 9, caspase 3, and β-actin in TTBS (25 mM Tris–HCl, 137 mM NaCl, 0.1% Tween 20, pH 7.4) containing 5% skim milk for 2 h. The membranes were then washed with TTBS and incubated with secondary antibodies. Signals from secondary antibodies were detected using the enhanced chemiluminescence (ECL) western blotting detection kit (Bio-Rad) and the image was captured using X-ray films. Relative protein expression was quantified by densitometric means using Multi Gauge v3.1 (FujiFilm, Tokyo, Japan) and calculated according to the reference β-actin bands. 

### 2.10. Flow Cytometric Assessment of Apoptosis 

Cell death by apoptosis was quantified in 5.5 mM and 30 mM glucose treatment cells, with or without HM-chromanone, using annexin-V/propidium iodide (PI) staining. A cell suspension (100 μL) was incubated with 5 μL of annexin-V and 5 μL of PI. This mixture was kept in the dark for 15 min in an ice bath. Thereafter, 400 μL of binding buffer was added to the cell suspension, which was then promptly subjected to flow cytometry (FACScalibur and CellQuest software; Becton Dickinson, San Jose, CA, USA). Apoptotic cells were expressed as a percentage of the total number of cells. 

### 2.11. Statistical Analysis

Each experiment was performed in triplicate. The results were expressed as the mean ± standard deviation (SD). Statistical analysis was performed using SAS software (SAS Institute, Inc., Cary, NC, USA). The treatment groups were compared by one-way analysis of variance (ANOVA) followed by a *post hoc* Duncan’s multiple-range test. A *p*-value of less than 0.05 was considered statistically significant. 

## 3. Results

### 3.1. (E)-5-Hydroxy-7-Methoxy-3-(2′-Hydroxybenzyl)-4-Chromanone 

(*E*)-5-hydroxy-7-methoxy-3-(2′-hydroxybenzyl)-4-chromanone is an yellow gum with the following characteristics; IR (infrared ray) (NaCl): υ_max_ = 3400-3300, 2945, 2861, 1582, 1453, and 1021 cm^−1^ UV (ultraviolet) (MeOH) λ_max_ 204, 215, and 281 nm; ^1^H NMR (300 MHz, CDCl_3_): 3.81 (s), 5.18 (d), 5.89 (d), 6.05 (d), 6.87 (m), 6.89 (m), 7.06 (d), 7.26 (t), and 7.94 (brs). ^13^C NMR (300 MHz, CDCl_3_): 56.3 (C-OCH3), 69.0 (C-2), 94.7(C-8), 95.9 (C-6), 104.3 (C-4a), 116.6 (C-3′), 120.3 (C-5′), 122.6 (C-1′), 130.1 (C-3), 131.5 (C-6′), 132.4 (C-4′), 134.7 (C-9), 157.8 (C-2′), 163.9 (C-8a), 166.0 (C-5), 169.3 (C-7), and 186.9 (C-4); LREIMS m/z 300.0998 [M]+1, C_17_H_14_O_5_. Purity of HM-chromanone was 100% (Figure 1). 

### 3.2. Effect of HM-Chromanone on Cell Viability

The effect of HM-chromanone on cell viability in INS-1 pancreatic β cells was assayed using the MTT assay. The cell viability was significantly reduced to 44.22% after treatment with 30 mM glucose in INS-1 pancreatic β cells, compared to the normal glucose treated cells (Figure 2). However, HM-chromanone protected against cell damage caused by high glucose in a dose-dependent manner. After HM-chromanone was added to the cells, cell damage was reversed and cell survival increased significantly to 54.09%, 70.42%, and 80.05% at the concentrations of 5, 10, and 20 µM, respectively, compared to the control group. These results suggest that HM-chromanone protects INS-1 pancreatic β cells from the damage induced by high glucose. 

### 3.3. Effect of HM-Chromanone on Intracellular Levels of Reactive Oxygen Species (ROS)

As shown in Figure 3, the generation of intracellular ROS in INS-1 pancreatic β cells was elevated significantly to 230.76% after treatment with high glucose compared to cells treated with 5.5 mM normal glucose. However, 1–20 µM HM-chromanone treatment dose-dependently decreased the levels of ROS in cells induced by 30 mM glucose. INS-1 pancreatic β cells treated with 20 µM HM-chromanone after high glucose pretreatment resulted in a significant decrease in ROS generation to 119.96%. Therefore, HM-chromanone significantly reduced high-glucose-induced intracellular ROS in INS-1 pancreatic β cells. 

### 3.4. Effect of HM-Chromanone on Generation of Thiobarbituric Acid Reactive Substances (TBARS)

As shown in Figure 4, the levels of TBARS induced with 30 mM glucose in INS-1 pancreatic β cells was significantly increased compared to the control group induced with 5.5 mM glucose. When INS-1 pancreatic β cells were exposed to 30 mM glucose for 48 h, TBARS were significantly increased to 0.33 nmol/MDA compared to the 0.17 nmol/MDA treated with 5.5 mM glucose (Figure 4). Treatment with 1, 5, 10, and 20 µM HM-chromanone significantly inhibited TBARS formation to 0.31, 0.29, 0.24, and 0.22 nmol MDA/mg protein, respectively, indicating protection against lipid peroxidation. Therefore, HM-chromanone significantly decreased the TBARS levels induced by high glucose treatment in INS-1 pancreatic β cells.

### 3.5. Effect of HM-Chromanone on the Level of Nitric Oxide (NO) 

The level of NO in INS-1 pancreatic β cells was significantly elevated by the treatment with 30 mM glucose, compared with 5.5 mM glucose-treated cells (Figure 5). However, the NO levels in HM-chromanone-treated cells were decreased, and this effect was concentration-dependent. The NO level was 223.10% in the INS-1 pancreatic β cells treated with high glucose, but treatment with 20 µM HM-chromanone resulted in a significant reduction in the NO levels, to 124.73%. These results implied that HM-chromanone could prevent the generation of NO by high glucose. 

### 3.6. Effect of HM-Chromanone on Insulin Secretion 

Figure 6 shows that the insulin secretion in INS-1 pancreatic β cells was significantly decreased as a result of high glucose (30 mM) treatment, compared to normal glucose (5.5 mM) treatment. After treating with 30 mM glucose and washing, 5 or 25 mM glucose were treated to analyze the insulin secretion ability. Treatment with HM-chromanone recovered 30 mM glucose-induced impairment of insulin secretion at 5 or 25 mM glucose. Treatment of INS-1 pancreatic β cells with 1, 5, 10, and 20 μM HM-chromanone increased the level of insulin secretion to 9.61, 10.88, 17.10, and 27.15 ng/h, respectively at 5 mM glucose (*p* < 0.05). In addition, HM-chromanone treatment increased insulin secretion to 12.93, 22.17, 23.61, and 33.98 ng/h in a dose-dependent manner at 25 mM glucose (*p* < 0.05). These results demonstrated that HM-chromanone increases insulin secretion in high glucose pretreated INS-1 pancreatic β cells.

### 3.7. Effect of HM-Chromanone on Apoptosis-Related Protein Expression 

To investigate whether HM-chromanone could suppress the expression of genes associated with high glucose-induced apoptosis, the protein levels of Bax, Bcl-2, cytochrome c, caspase 9, and caspase 3 were measured after treating INS-1 pancreatic β cells pretreated with high glucose with 10 or 20 µM HM-chromanone. Figure 7 show that the levels of Bax, cytochrome C, caspase 9, and caspase 3 protein expression were significantly increased, whereas Bcl-2 expression levels significantly decreased in high glucose-treated INS-1 pancreatic β cells. However, the treatments with 10 or 20 μM HM-chromanone significantly decreased Bax expression in high glucose-pretreated INS-1 pancreatic β cells, and also significantly decreased the expression levels of caspase 9, caspase 3, and cytochrome *C*. On the contrary, treatment HM-chromanone significantly increased the expression level of Bcl-2 protein in high glucose pretreated cells. These results implied that the HM-chromanone treatment reduced pro-apoptotic protein expression and increased anti-apoptotic protein expression in the β cells damaged by high glucose, and protected the cells from apoptosis.

### 3.8. Identification of the Type of Cell Death By Annexin-V/PI Staining

The flow cytometry analysis with annexin-V/PI staining was used to determine the rate of apoptosis. The bottom and top of right quadrant areas indicate the number of early apoptotic and late apoptotic cells (Figure 8). When INS-1 pancreatic β cells were treated with high glucose (30 mM), the number of apoptosis cells was higher than that in cells treated with normal glucose (5.5 mM). However, HM-chromanone treatment reduced the number of apoptosis cells in 30 mM glucose-pretreated cells in a dose-dependent manner (Figure 8Ac,d, lower and upper right quadrants). In particular, treatment with 20 μM HM-chromanone resulted in a significantly decreased number of apoptosis cells. Here, only 37.93% of cells were viable when the cells were treated with 30 mM high glucose alone, but the percentage of viable cells increased to 76.30% after treatment with 20 μM HM-chromanone. These findings demonstrated that HM-chromanone protected these INS-1 pancreatic β cells from apoptosis induced by exposure to a high glucose level. 

## 4. Discussion

Hyperglycemia causes many of the pathological consequences of type 2 diabetes [3]. Most of these pathological impairments are a result of increased production of ROS. The oxidative stress caused by ROS has recently been reported as an integrative factor for the impairment of hyperglycemia [6]. In particular, pancreatic β-cells are extremely sensitive towards oxidative stress. The damage of β-cells by hyperglycemia-induced oxidative stress ultimately leads to apoptotic cell death and reduces insulin secretion [8]. In order to alleviate this damage, it is important to reduce the oxidative stress caused by high glucose [18]. HM-chromanone, a natural homoisoflavonoid component, was isolated from *P. oleracea*. Homogeneous isoflavonoids exhibit biological activity associated with oxidative stress, such as antioxidant and anti-inflammatory activity [19]. Thus, this study examined whether HM-chromanone could protect INS-1 β cells from oxidative stress and apoptosis induced by high glucose. 

The effect of HM-chromanone on cell viability in INS-1 pancreatic β cells was first examined using the MTT assay. The exposure of the β cells to the high glucose levels significantly decreased their cell viability, but HM-chromanone treatment recovered their survival rate. This result indicated that HM-chromanone protected the INS-1 pancreatic β cells from cytotoxicity caused by high glucose levels.

In diabetes, hyperglycemia produces excess reactive oxygen species (ROS), which can result in a variety of biochemical and physiological lesions. Since pancreatic β cells have low antioxidative defense properties, ROS amassment induces the activation of stress-sensitive intracellular signaling pathways, promoting cellular damage and contributing to chronic diabetic complications and progression [20,21]. Thus, the inhibition of ROS production plays a vital role in mitigating the hyperglycemia-induced damage in pancreatic β cells [22]. The present study demonstrated that the treatment of high glucose in INS-1 pancreatic β cells significantly increased intracellular ROS generation. However, HM-chromanone treatment significantly decreased the generation of ROS induced by high levels of glucose in the cells. Previous studies have shown that homoisoflavonoids isolated from natural plants have scavenging activity and are able to remove free radicals. Calvo reported that three homoisoflavanones were isolated from the bulbs of *Ledebouria floribunda*. These compounds exhibited potent antioxidant activity in the 2,2-diphenyl-1-picrylhydrazyl (DPPH) radical-scavenging [23]. Homoisoflavonoids were also isolated from *Caesalpinia sappan* and significantly attenuated intracellular ROS production [24].

Homoisoflavonoids belong to a type of special flavonoids. Its general structure is a 16-carbon skeleton consisting of two phenyl rings (A and B) and a heterocyclic ring (C) [25]. Their B- and C-rings were connected by an oxygenated substituent (hydroxyl, methoxyl, or bridged with –OCH2O–) [26]. It is known that OH substitution is necessary for the antioxidant activity of a flavonoid [27]. The flavonoids that contain multiple OH substitutions have very strong antioxidant activities against peroxyl radicals [28]. Thus, the scavenging effects of HM-chromanone on ROS might be attributable to the OH substitutions in the structure.

Increased oxidative stress is the consequence of either enhanced ROS production or attenuated ROS scavenging capacity, resulting in cell damage that is assessed by the measurement of lipid peroxides. Thus, the measurement of lipid peroxidation could provide a good indicator of cell impairment of INS-1 pancreatic β cells [29,30]. The exposure of INS-1 pancreatic β cells to high glucose levels increased the generation of lipid peroxides; however, HM-chromanone treatment significantly inhibited lipid peroxide generation. This result suggests that the inhibition of lipid peroxide formation may be one of the protective mechanisms involved in HM-chromanone mediated reduction of high glucose-induced cytotoxicity in INS-1 pancreatic β cells. Lipid peroxidation is caused by ROS attacking the cells and can induce β-cell dysfunction and reduce insulin secretion [31]. Therefore, the production of TBARS should be prevented and β-cell malfunction by oxidative stress should be reduced. Recent studies have shown that flavonoids inhibit TBARS production due to antioxidant action [32,33]. The structure of flavonoids is an important determinant for radical scavenging and for inhibiting lipid peroxidation. These structures are the 4-oxo functional group of the B ring and the 3- and 5-hydroxyl groups in the A ring [34]. The efficiency of HM-chromanone in inhibiting lipid peroxidation is partially consistent with these structures. 

Diabetes is a metabolic disorder characterized by chronic hyperglycemia [35]. Increased glucose levels over the long-term were associated with high nitric oxide (NO) production [36]. Low levels of NO are beneficial to a variety of physiological and cellular functions, but high levels of NO can have deleterious effects on β cells [37]. Furthermore, NO leads to highly reactive oxidative damage associated with diabetes [38]. In this study, the increase in NO levels induced by high glucose in INS-1 pancreatic β cells was significantly reduced by HM-chromanone. These results suggest that HM-chromanone can significantly decrease the levels of NO produced in high glucose environments and protect pancreatic β cells from oxidative damage. 

Oxidative stress activates apoptosis signal transduction pathways in many cell types [39]. It also controls processes involved in the initiation of apoptotic signaling, performing pro-apoptotic roles. Mitochondria are known to induce apoptosis, releasing cytochrome *c* into the cytosol, which leads to the assembly of a caspase-activating complex referred to as the “apoptosome” [40]. Bax is a pro-apoptotic protein that can promote apoptosis induced by mitochondrial membrane permeability, resulting in the release of apoptogenic factors from the mitochondria. On the other hand, Bcl-2 is an anti-apoptotic protein that can preserve the integrity of the outer membrane of mitochondria [41]. Changes in the concentrations of these gene products can further stimulate apoptotic events, including changes in the mitochondria that ultimately lead to the activation of a family of cysteine proteases called caspases [42]. Among these, caspase 3 and caspase 9 are frequently activated death proteases, catalyzing the specific cleavage of many key cellular proteins [43]. In this study, high glucose levels induced apoptosis via the activation of the mitochondrial apoptotic pathway in INS-1 pancreatic β cells. However, HM-chromanone treatment markedly reduced Bax expression, increased the level of Bcl-2 expression, and significantly decreased the activation of the cytochrome c, caspase 3, and caspase 9. 

In some studies, the hydroxyl groups of a compound isolated from seaweed and plants have been shown to exhibit anti-apoptotic effects and are involved in the regulation of mitochondrial apoptotic pathways [44,45]. The anti-apoptotic effect of the hydroxyl group is related to the inhibition of intracellular ROS and lipid peroxidation. The hydroxyl group in the C-5 of food-derived polyphenols especially helps to scavenge generated free radicals and is considered contributory for its antioxidant activity [46,47]. Polyphenols may modulate the mitochondrial membrane and maintain the levels of apoptotic and anti-apoptotic proteins as a result of the presence of a hydroxyl group [47]. HM-chromanone has two hydroxyl groups at C-5 and C-2′ position. Thus, we assume that these two hydroxyl groups might significantly contribute to improving the anti-apoptotic effect in INS-1 pancreatic β cells. This study suggests that HM-chromanone protected INS-1 pancreatic β cells from the apoptosis induced by high levels of glucose. This effect might be explained by increased anti-apoptotic Bcl-2 expression and decreased pro-apoptotic Bax, cytochrome c, caspase 9, and caspase 3 expressions. 

Flow cytometry measurement was conducted in order to investigate the rate of apoptosis. Annexin-V was used as a cell death marker. The lower and upper right quadrant regions represent the number of early apoptotic and late apoptotic cells. Exposure to a high glucose level increased the rate of apoptosis in INS-1 pancreatic β cells [48]. However, HM-chromanone treatment reduced the rate of apoptosis in the high glucose pretreated cells. The number of apoptotic cells was markedly decreased after treatment with 20 μM HM-chromanone compared to the number of apoptotic cells in high-dose glucose-treated cells. This result suggests that HM-chromanone protects INS-1 pancreatic β cells during apoptosis induced by high glucose. 

Pancreatic β cells dysfunction is associated with insulin-secreting defects due to glucose stimulation [49]. It has been reported that increase in glucose from 5.5 to 30 mM in humans and rat islets leads to a definite increase in beta cell apoptosis, which reduces insulin secretion [50,51]. In this study, insulin secretion was significantly reduced when INS-1 pancreatic β cells were incubated with 30 mM glucose; however, insulin secretion was significantly increased dose-dependently with HM-chromanone treatment. This result suggests that HM-chromanone can prevent pancreatic β-cell dysfunction induced by hyperglycemia and restore the insulin secretion function in pancreatic β-cells.

## 5. Conclusions

In conclusion, exposure of INS-1 pancreatic β cells to a high level of glucose lead to apoptosis by increasing the levels of reactive oxygen species, lipid peroxides, and NO. As a result, the level of insulin secretion decreased. However, HM-chromanone treatment significantly decreased the levels of intracellular ROS, lipid peroxide, and NO. This treatment also significantly enhanced anti-apoptotic Bcl-2 expression and significantly decreased the activation of Bax, cytochrome c, caspase 3, and caspase 9. Both insulin secretion and cell viability increased in high glucose pretreated INS-1 pancreatic β cells after treatment with HM-chromanone. These results suggest that HM-chromanone could be useful as a pharmaceutical agent for the protection of pancreatic β-cells against oxidative stress and apoptosis induced by high glucose. 

## Figures and Tables

**Figure 1 nutrients-11-00404-f001:**
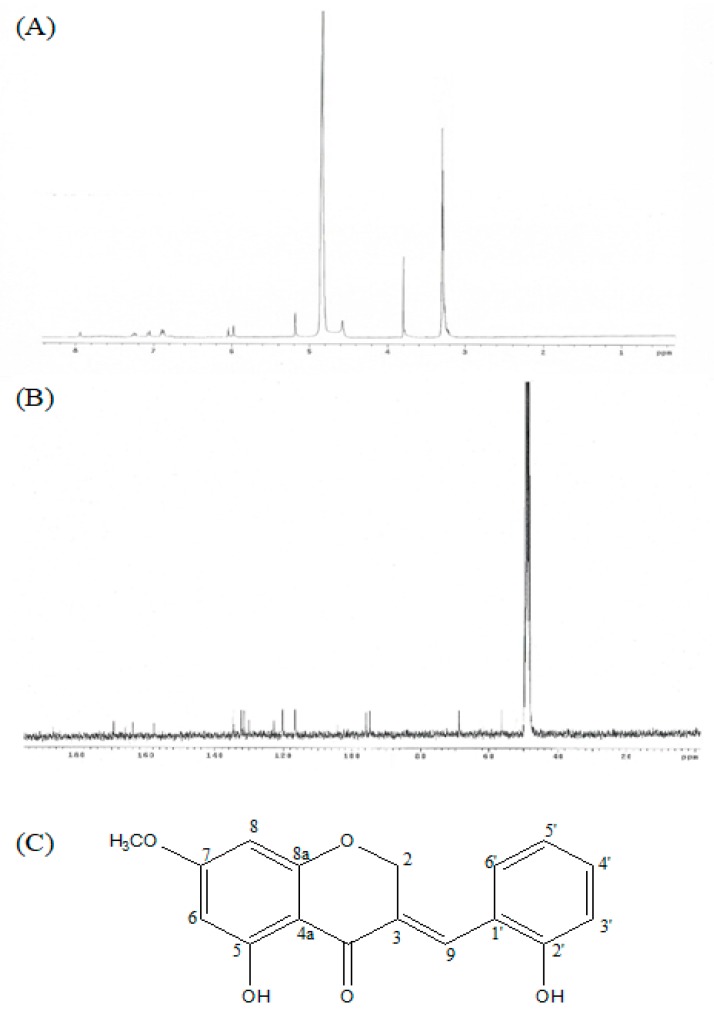
^1^H NMR, ^13^C NMR, and chemical structure of HM-Chromanone isolated from *P. oleracea*. ^1^H NMR, ^13^C NMR, and chemical structure of HM-Chromanone isolated from *P. oleracea*. (**A**) ^1^H NMR. (**B**) ^13^C NMR spectrum, and (**C**) chemical structure of (*E*)-5-hydroxy-7-methoxy-3-(2′-hydroxybenzyl)-4-chromanone (HM-chromanone).

**Figure 2 nutrients-11-00404-f002:**
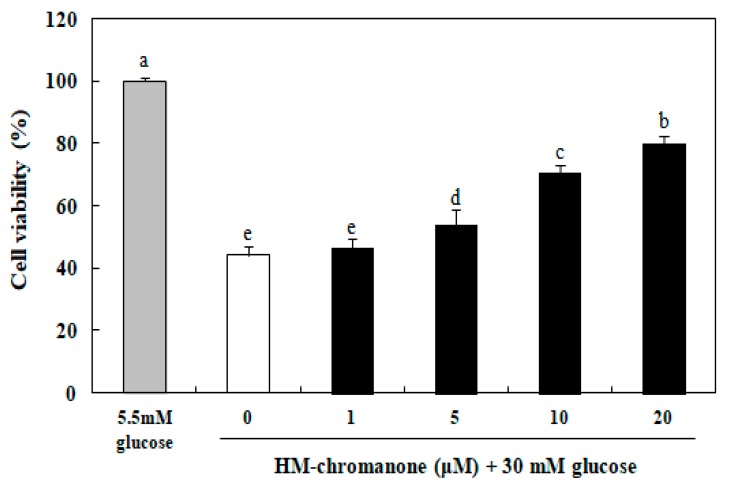
Effect of HM-chromanone on cell viability in high glucose-treated INS-1 pancreatic β cells. INS-1 pancreatic β cells (2 × 10^4^ cells/well) were preincubated in 96-well plates with 5.5 or 30 mM glucose for 48 h, and then incubated with HM-chromanone (0, 1, 5, 10, or 20 µM) for 48 h. The 5.5 mM glucose represents normal glucose, while the 30 mM glucose represents high glucose concentrations. Each value is expressed as the mean ± standard deviation (SD) (*n* = 3). ^a~e^ Values with different letters were significantly different at *p* < 0.05, as analyzed by Duncan’s multiple-range test.

**Figure 3 nutrients-11-00404-f003:**
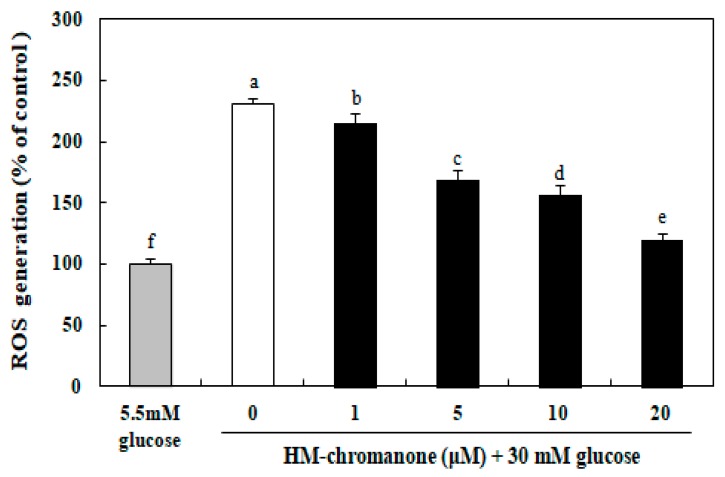
Effect of HM-chromanone on intracellular levels of reactive oxygen species (ROS) in high glucose-treated INS-1 pancreatic β cells. INS-1 pancreatic β cells (2 × 10^4^ cells/well) were preincubated with 5.5 or 30 mM glucose in 96-well plates for 48 h, and then incubated with HM-chromanone (0, 1, 5, 10, or 20 µM) for 48 h. The concentration of 5.5 mM glucose represents normal glucose, while the 30 mM glucose represents a high glucose concentration. Each value is expressed as the mean ± standard deviation (*n* = 3). ^a~f^ Values with different letters were significantly different at *p* < 0.05, as analyzed by Duncan’s multiple-range test.

**Figure 4 nutrients-11-00404-f004:**
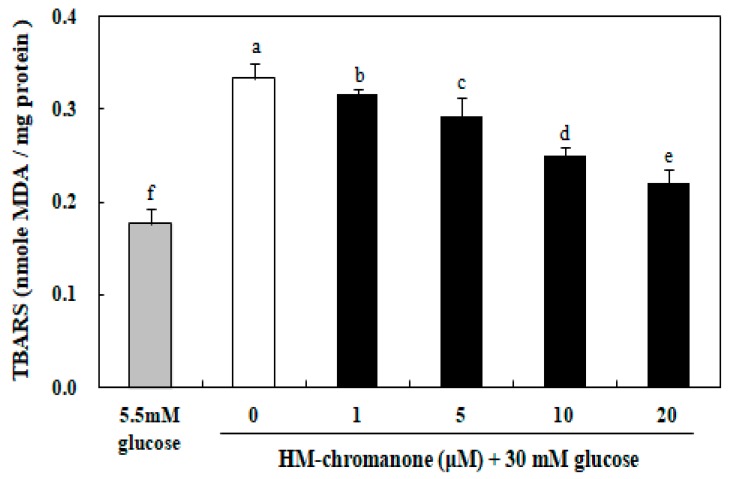
Effect of HM-chromanone on the generation of thiobarbituric acid reactive substances (TBARS) in high glucose-treated INS-1 pancreatic β cells. INS-1 pancreatic β cells (2 × 10^4^ cells/well) were preincubated in 96-well plates with 5.5 or 30 mM glucose for 48 h, and then incubated with HM-chromanone (0, 1, 5, 10, or 20 µM) for 48 h. The concentration of 5.5 mM glucose represents normal glucose, while 30 mM glucose represents a high glucose concentration. Each value is expressed as the mean ± standard deviation (*n* = 3). ^a~f^ Values with different letters were significantly different at *p* < 0.05, as analyzed by Duncan’s multiple-range test.

**Figure 5 nutrients-11-00404-f005:**
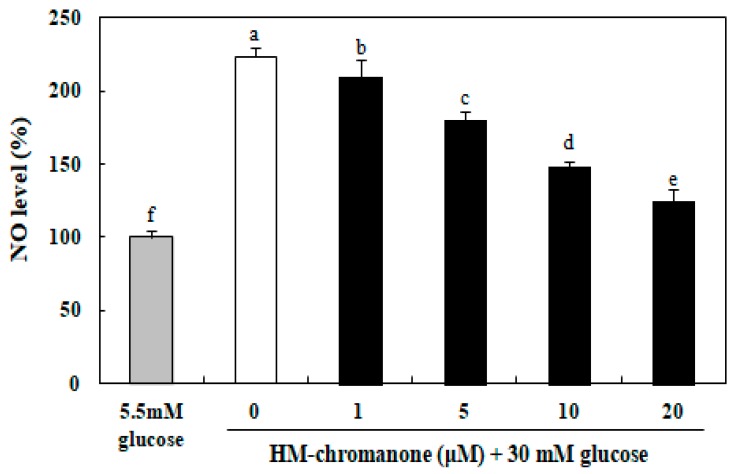
Effect of HM-chromanone on the level of nitric oxide (NO) in high glucose-treated INS-1 pancreatic β cells. INS-1 pancreatic β cells (2 × 10^4^ cells/well) were preincubated in 96-well plates with 5.5 or 30 mM glucose for 48 h, and then incubated with HM-chromanone (0, 1, 5, 10, or 20 µM) for 48 h. The concentration of 5.5 mM glucose represents normal glucose, while 30 mM glucose represents a high glucose concentration. Each value is expressed as the mean ± standard deviation (*n* = 3). ^a~f^ Values with different letters were significantly different at *p* < 0.05, as analyzed by Duncan’s multiple-range test.

**Figure 6 nutrients-11-00404-f006:**
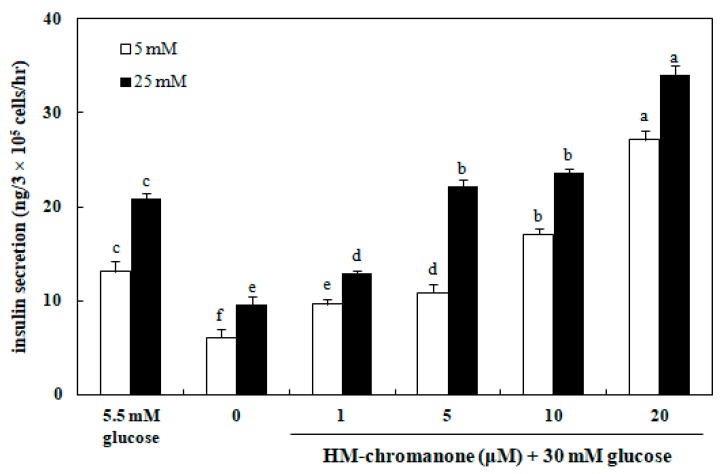
Effect of HM-chromanone on insulin secretion in high glucose-treated INS-1 pancreatic β cells. INS-1 pancreatic β cells (2 × 10^5^ cells/well) were preincubated in 96-well plates with 5.5 or 30 mM glucose, and then incubated with HM-chromanone (0, 1, 5, 10, or 20 µM) for 48 h. Thereafter, the cells were stimulated with Krebs–Ringer buffer containing 5 or 25 mM glucose for 60 min. The 5.5 mM concentration of glucose represents normal glucose, while 30 mM glucose represents a high glucose concentration. Each value is expressed as the mean ± standard deviation (*n* = 3). ^a~f^ Values with different letters were significantly different at *p* < 0.05, as analyzed by Duncan’s multiple-range test.

**Figure 7 nutrients-11-00404-f007:**
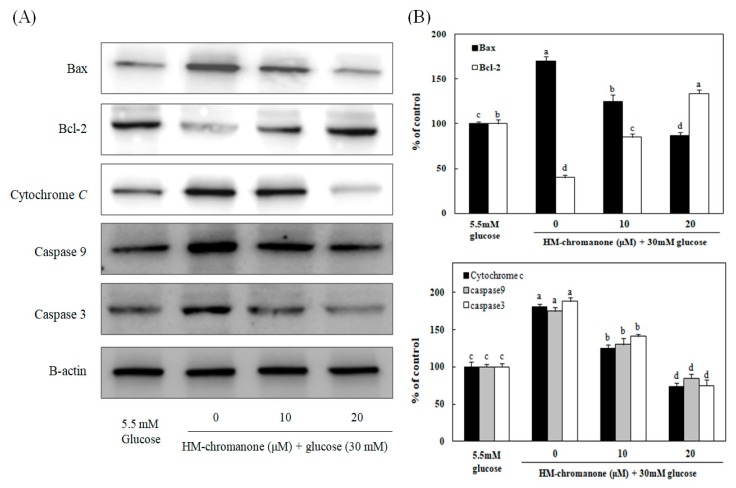
Effect of HM-chromanone on expressions of Bax, Bcl-2, cytochrome C, caspase 9, and caspase 3 in high glucose-treated INS-1 pancreatic β cells. INS-1 pancreatic β cells were preincubated with 5.5 or 30 mM glucose for 48 h, and then incubated with HM-chromanone (10 or 20 µM) for 48 h. Equal amounts of cell lysates were electrophoresed and levels of Bax, Bcl-2, cytochrome C, caspase 9, and caspase 3 protein expression were measured using western blots. (**A**) Bax, Bcl-2, cytochrome C, caspase 9, and caspase 3 protein levels. (**B**) Expression levels of Bax, Bcl-2, cytochrome C, caspase 9, and caspase 3. Each value is expressed as the mean ± standard deviation (*n* = 3). ^a~d^ Values with different letters were significantly different at *p* < 0.05, as analyzed by Duncan’s multiple-range test.

**Figure 8 nutrients-11-00404-f008:**
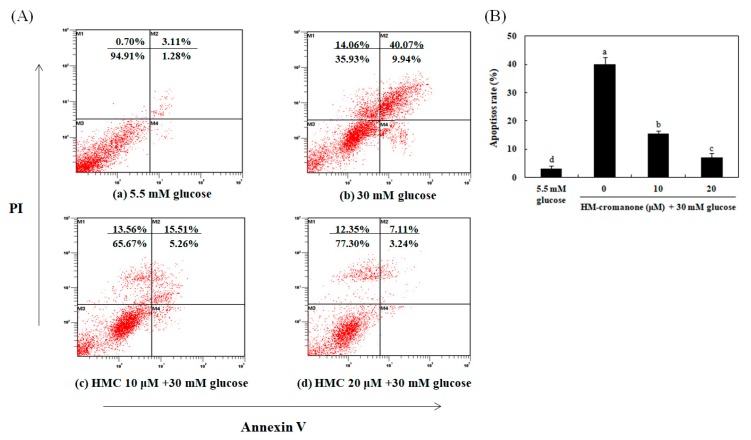
Identification of the type of cell death by Annexin V-FITC/propidium iodide (PI) staining. The status of apoptotic cell death was determined by counting INS-1 pancreatic β cells stained with annexin V-FITC/PI using a flow cytometer. Cells were preincubated with glucose and then incubated in the presence or absence of HM-chromanone (10 or 20 µM). The lower and upper right quadrants show the numbers of early and late apoptotic cells, respectively.

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
