# Peer review of "HM-Chromanone Isolated from Portulaca Oleracea L. Protects INS-1 Pancreatic β Cells against Glucotoxicity-Induced Apoptosis"

_nutrients, 2019, doi:10.3390/nu11020404_

Round 1
Reviewer 1 Report
The paper by Park et al. describes the effect of HM-chromanone extracted from Portulaca oleacea L. in protecting INS-1 cells from glucotoxicity-induced apoptosis.
The strategy addresses an important aspect of beta-cell biology and is relevant for the development of novel therapy for treatment of Diabetes.
The technical approach appears adequate for the scope of the project but is opinion of the reviewer that there are some aspects the authors should address.
1) It would be useful to know the purity of the HM-chromanone.
2) The INS-1 media reported is not correct. It is missing 50 uM beta-mercaptoethanol.
3) The insulin secretion experiments reported do not show the results of a real GSIS. First, the methods section describes a GSIS but no mention of glucose stimulation is reported with the KRBH treatment. The experiments should be repeated reporting for each condition insulin secretion values determined in low (2.8 mM) Glucose and high (16.7 mM) Glucose. Furthermore, the results should be normalized for the number of cells for each condition to take in account the reduction of cell due apoptosis (for the 0 uM HM-chromanone) and the progressive rescue due to the effect of the HM-chromanone
4) In Figure 3, the label of the Y-axis (%) is misleading. It would be more appropriate to report the absolute value as Relative Fluorescence Units (RFU).
5) In Figure 7B the Y-axes should be labeled.
6) In row 337, the reference (Rosshnak Namdar, 2013) should be reported as a number.
7) In row 448, the title of the citation is incomplete.
Author Response
Response to reviewers
I pick up the check and look at your comments and checklist.
And I have corrected everything you have pointed out.
Please review my paper and reply to e-mail if you would like to make any corrections.
Thank you.

Reviewer 2 Report
This study addresses the potential beneficial effects of HM-chromanone isolated from P. oleracea L. in protecting pancreatic beta cells against gluco-toxicity induced apoptosis.
For the study, HM-chromanone was extracted and purified, and varying doses were administered to INS-1 cells in culture. The addition of increasing concentrations of HM-chromanone induced a dose-dependent effect in reducing ROS, TBARs and NO production in a model of glucotoxicity (cells exposed to 30 mM glucose under in-vitro conditions) while increasing insulin secretion. Assessment of apoptosis indicates that the agent reduced the expression of Bax, cyt-C, Caspase 3 and Caspase 9 while increasing Bcl-2 expression. Overall, the conclusion of the authors is that HM-chromanone elicits a dose-dependent protection on the toxicity induced by exposure of INS-1 beta cells to high doses of glucose.
The manuscript is properly written.
Comments:
The authors have provided data consistent with HM-chromanone increasing insulin secretion by decreasing ROS-medicated apoptosis and cell viability under hyperglycemic conditions. However, an additional possibility could be that HM-chromanone increases glycolysis-related ATP production by improving mitochondria activity. Do the authors have any indication whether HM-chromanone modulates glucose entry and the associated ATP production within INS-1 cells? If the agent is administered to INS-1 cells in the presence of a physiological glucose level is there any increase in the amount of ATP produced and insulin outputted by the cells? The protective effect of HM-chromanone could also depend on an improved glycolysis/ATP production in the beta cells
Author Response

(The authors gave the same response as above.)

Round 2
Reviewer 2 Report
In this manuscript, the authors investigate the potential beneficial effects of HM-chromanone isolated from P. oleracea L. in protecting pancreatic beta cells against gluco-toxicity induced apoptosis. For the study, HM-chromanone was extracted and purified, and varying doses were administered to INS-1 cells in culture. The addition of increasing concentrations of HM-chromanone induced a dose-dependent effect in reducing ROS, TBARs and NO production in a model of glucotoxicity (cells exposed to 30 mM glucose under in-vitro conditions) while increasing insulin secretion. Assessment of apoptosis indicates that the agent reduced the expression of Bax, cyt-C, Caspase 3 and Caspase 9 while increasing Bcl-2 expression. Overall, the conclusion of the authors is that HM-chromanone elicits a dose-dependent protection on the toxicity induced by exposure of INS-1 beta cells to high doses of glucose.
The manuscript is properly written.